# Responses of Growth, Oxidative Injury and Chloroplast Ultrastructure in Leaves of *Lolium perenne* and *Festuca arundinacea* to Elevated O_3_ Concentrations

**DOI:** 10.3390/ijms23095153

**Published:** 2022-05-05

**Authors:** Sheng Xu, Yan Li, Bo Li, Xingyuan He, Wei Chen, Kun Yan

**Affiliations:** 1CAS Key Laboratory of Forest Ecology and Management, Institute of Applied Ecology, Shenyang 110016, China; shengxu703@126.com (S.X.); lib@iae.ac.cn (B.L.); chenwei@iae.ac.cn (W.C.); 2College of Resources and Environment, University of Chinese Academy of Sciences, Beijing 100049, China; ly_qianer@163.com; 3Shenyang Arboretum, Chinese Academy of Sciences, Shenyang 110016, China; 4School of Agriculture, Ludong University, Yantai 264025, China

**Keywords:** oxidative stress, ozone exposure, reactive oxygen species, chloroplast, antioxidative enzymes, urban plants

## Abstract

The effects of increasing atmospheric ozone (O_3_) concentrations on cool-season plant species have been well studied, but little is known about the physiological responses of cool-season turfgrass species such as *Lolium perenne* and *Festuca arundinacea* exposed to short-term acute pollution with elevated O_3_ concentrations (80 ppb and 160 ppb, 9 h d^−1^) for 14 days, which are widely planted in urban areas of Northern China. The current study aimed to investigate and compare O_3_ sensitivity and differential changes in growth, oxidative injury, antioxidative enzyme activities, and chloroplast ultrastructure between the two turf-type plant species. The results showed that O_3_ decreased significantly biomass regardless of plant species. Under 160 ppb O_3_, total biomass of *L. perenne* and *F. arundinacea* significantly decreased by 55.3% and 47.8% (*p* < 0.05), respectively. No significant changes were found in visible injury and photosynthetic pigment contents in leaves of the two grass species exposed to 80 ppb O_3_, except for 160 ppb O_3_. However, both 80 ppb and 160 ppb O_3_ exposure induced heavily oxidative stress by high accumulation of malondialdehyde and reactive oxygen species in leaves and damage in chloroplast ultrastructure regardless of plant species. Elevated O_3_ concentration (80 ppb) increased significantly the activities of superoxide dismutase, catalase and peroxidaseby 77.8%, 1.14-foil and 34.3% in *L. perenne* leaves, and 19.2%, 78.4% and 1.72-fold in *F. arundinacea* leaves, respectively. These results showed that *F. arundinacea* showed higher O_3_ tolerance than *L. perenne*. The damage extent by elevated O_3_ concentrations could be underestimated only by evaluating foliar injury or chlorophyll content without considering the internal physiological changes, especially in chloroplast ultrastructure and ROS accumulation.

## 1. Introduction

The concentration of global atmospheric ozone (O_3_) is increasing by 1–2% per year from 10 ppb in the 1900s to the current level of 40 ppb [1]. Furthermore, the atmospheric O_3_ concentration is predicted to reach 70 ppb by the year 2050, while the regional spikes as high as 200 ppb have become rather frequent, especially in summer [2]. The ground-level O_3_ concentrations have significantly increased in recent decades in many regions including Shenyang city of Northeast China in this study [3,4,5]. Acute exposure of elevated O_3_ concentration within a short time period often occurred during the growing season of plants in many cities of China [5,6,7].

As one of the most toxic air pollutants, elevated O_3_ concentrations could exert serious consequences on plant growth, physiological metabolisms and morphological characteristics [8,9,10,11,12]. The adverse effects of elevated O_3_ concentrations on plants including herb species have been well known [13,14]. Elevated O_3_ concentration led to leaf injury, decreased photosynthetic rate, and inhibited growth and accelerated senescence of many plant species [15,16,17,18]. In particular, O_3_ entered the plants through stomata and it was degraded into reactive oxygen species (ROS) [19]. Excess ROS could make plant metabolic disorder by causing irreversible damage to physiological processes and cell structure such as chloroplast ultrastructure [20]. Elevated O_3_ concentrations (more than 80 ppb) could cause acute damage on plants and chronic injuries [21]. However, the responses of plants to O_3_ are species specific [22]. Some plant species adapted well to O_3_ environments [23], but some were more sensitive and could be used as bio-indicators for O_3_ pollution [24]. Actually, plants including herb species have developed antioxidative systems to protect plant cells by regulating the production rate or content of intra-cellular ROS such as hydrogen peroxide (H_2_O_2_) and superoxide anion (O_2_**·**^−^). Changes in the activities of defense enzymes in leaves such as superoxide dismutase (SOD, catalase (CAT), peroxidase (POD) and ascorbate peroxidase (APX) could play an important role in preventing and alleviating the adverse impacts of elevated O_3_ by scavenging ROS in many plants [9,14,17,20].

Perennial ryegrass (*Lolium perenne* cv. Lark) and tall fescue (*Festuca arundinacea* cv. Pixie) used in this study are two improved turf-type species in cool-season plants, and applied widely in the cities of temperate or cool climate areas for many years in China [25]. As cool-season plant species, both of them were more vulnerable to high temperature or heat stress, especially in summer in urban areas or many cities of China, and perennial ryegrass was more sensitive to high temperature than tall fescue [26]. In addition, the two grass species have excellent characteristics such as easy to establish, improved drought and low temperature adaptation and less-cost maintenance in establishing turf or lawn compared to common cultivars [25], but they may also be easy to suffer the adverse impact or stress from air pollution such as elevated O_3_ concentrations occurred often in dry summer. Many physiological changes could be involved in O_3_ stressfor cool-season herb species [13,14,24,27]. However, little is known about how elevated O_3_ concentrations impact the changes of ROS and chlorophyll ultrastructure of cool-season turfgrass species. In gerenal, *L. perenne* is more sensitive to abiotic stresses than *F. arundinacea* [13,25,26]. Therefore, we here hypothesized that *L. perenne* could be more sensitive to O_3_ than *F. arundinacea* based on the comparison of the physiological changes between them. The main purposes of this study were to (1) compare and evaluate the differences and extents of oxidative injury and the antioxidant protection suffering from O_3_ stress; (2) to determine whether and to what extent chloroplasts in leaves of the two cool-season grass species would be affected by the short-term acute fumigation of elevated O_3_ concentrations, which occurs frequently in summer under natural condition in the urban area.

## 2. Results

### 2.1. Visible Injury, Pigment Contents and Chloroplast Ultrastructure

No visible foliar injury was observed in leaves of *L. perenne* and *F. arundinacea* exposed to AA and 80 ppb O_3_ concentration (Figure 1). Ozone-induced visible foliar injury was observed on the two plant species exposed to 160 ppb O_3_ concentration (Figure 1). Foliar injury symptoms on the leaves of the two grass species were similar and characterized as the diffuse chlorotic stripes and brown necrosis appeared first in leaf tip.

To evaluate the chlorosis degree and compare the changes in the contents of photosynthetic pigmentsincluding chlorophyll (Chl) and carotenoids (Car) in leaves of *L. perenne* and *F. arundinacea* were shown in Figure 2. In contrast, Chl a content decreased by elevated O_3_ (160 ppb) in *L. perenne* leaves by about 29.1% as compared to ambient air (AA), and in *F. arundinacea* leaves by 16.5% (*p* < 0.05). No significant effect between different treatments on Chl b was found in *F. arundinacea* leaves. Chlorophyll a + b content showed significant decrease under elevated O_3_ concentration (160 ppb), particularly for *L. perenne* leaves (by 23.5%, *p* < 0.05). Chl a + b and Car contents had not significant change in plant leaves exposed to 80 ppb O_3_ regardless of grass species, but decreased significantly under elevated O_3_ (160 ppb, *p* < 0.05). GLM analysis showed that no significant interactive effect of O_3_ and species was found on Chl a + b (*p* = 0.074, Table 1) and Car contents (*p* = 0.875, Table 1).

To identify morphological changes in photosynthetic organelles in leaves of *L. perenne* and *F. arundinacea*, the chloroplast ultrastructure was examined. As shown in Figure 3, the chloroplasts of the two plant species under ambient air exhibited an elliptical shape, with a typical lamellar grana structure consisting of thylakoid and several large starch grains and few osmiophilic granules, especially for *F. arundinacea* (Figure 3A,D). By contrast, the chloroplasts in leaves exposed to elevated O_3_ concentrations were abnormal and swollen in appearance (Figure 3B,F), with lots of osmiophilic granules and a serious separation of chloroplast from cell wall (Figure 3B,C,E,F). In 160-ppb O_3_ exposed leaves, chloroplasts were partly damaged with an irregular and slack grana thylakoid and completely separated from cell wall, and mitochondria was degraded with a part disappear of inner wrinkle, especially for *L. perenne* (Figure 3C).

### 2.2. Oxidative Injury

The oxidative injury induced by elevated O_3_ concentrations was shown in Figure 4. Compared with the plants under AA, the O_3_-exposed leaves of both *L. perenne* and *F. arundinacea* show higher MDA content, O_2_**·**^−^ production rate and H_2_O_2_ content (Figure 3, *p* < 0.05). Elevated O_3_ concentration (80 ppb) increased significantly MDA content, O_2_**·** production rate and H_2_O_2_ content by 33.5%, 29.3% and 33.5% in *L. perenne* leaves (*p* < 0.05), − and 48.1%, 25.1% and 14.6% in *F. arundinacea* leaves (*p* < 0.05), respectively. MDA content, O_2_**·**^−^ production rate and H_2_O_2_ content significantly increased by 91.4%, 36.8% and 68.2% in leaves of *F. arundinacea* exposed to 160 ppb O_3_ concentration (*p* < 0.05, Figure 4). The increasing of O_2_**·**^−^ production rate in *L. perenne* leaves (68.9%) showed higher level than in *F. arundinacea* leaves (36.8%). GLM analysis showed a significant interactive effect of O_3_ and species on H_2_O_2_ content (*p* < 0.01), except for MDA content (*p* = 0.430, Table 1) and O_2_**·**^−^ production rate (*p* = 0.145, Table 1).

### 2.3. Changes in Antioxidative Enzyme Activities

Elevated O_3_ significantly increased the activities of SOD, CAT and POD, regardless of plant species (Figure 5). For *L. perenne*, SOD, CAT and POD activities increased by 77.8% and 1.14-fold, 34.3% and 91.7%, and 1.25-fold and 2.40 fold under 80 ppb and 160 ppb O_3_ concentrations (*p* < 0.05), respectively. In *F. arundinacea* leaves, SOD, CAT and POD activities increased by 19.2% and 47.1%, 78.4% and 1.73-fold, and 1.72-fold and 3.82 fold under 80 ppb and 160 ppb O_3_ concentrations (*p* < 0.05), respectively. For the two grass species, APX activity increased first under 80 ppb O_3_ and decreased under 160 ppb O_3_ (Figure 5). Increase of APX activity in leaves of *L. perenne* (by 68.7%) under 80 ppb O_3_ showed higher level than that under 160 ppb O_3_ (24.5%). Regardless of grass species, GLM analysis showed a significant interactive effect of O_3_ and species on antioxidative enzyme activities (*p* < 0.01, Table 1), except for APX activity (*p* = 0.473, Table 1).

### 2.4. Growth Response

Elevated O_3_ concentrations can decrease growth in many plant species. Growth parameters of *L. perenne* and *F. arundinacea* were shown in Figure 6. In this study, elevated O_3_ concentraton induced a significant reduction of growth in the two turfgrass species, especially in biomass. With increasing of O_3_ concentrations, aboveground biomass, root biomass and total biomass of the two grass species showed a trend of decreasing, respectively. Compared with AA, total biomass significantly decreased by 33.7% and 55.3% in *L. perenne* (*p* < 0.05), 42.9% and 47.8% in *F. arundinacea* (*p* < 0.05) exposed to 80 ppb and 160 ppb O_3_, respectively. No significant difference in total biomass was found between the two elevated O_3_ concentrations for each species, particularly in *F. arundinacea* (*p* = 0.597). No difference in R/S ratio was observed among the different factors (Figure 6). No significant interactive effect was found between O_3_ and species in all growth parameters (Table 1).

## 3. Discussion

Many studies showed that in general, elevated O_3_ concentration could induce typical visible injuries in plant leaves [28,29]. Consequently, these injuries were considered as the important traits for evaluating sensitivity or tolerance of different species or cultivars to O_3_ [28,30,31]. Actually, the formation and occurrence of visible injury symptoms might show retardation effect compared to internal physiological changes. In the present study, we examined two common turf-type herb species and their responds of growth, foliar visible injury, chlorophyll content, oxidative status and antioxidative ability to elevated O_3_ concentrations. Also, we performed a comparative analysis of anatomic characteristics for chloroplast ultrastructure in leaves of *L. perenne* and *F. arundinacea*. To our knowledge, this is the first study on physiological mechanisms underlying the sensitivity to elevated O_3_ concentrations in the two cool-season turfgrass species.

### 3.1. Visible Injury and Growth Characteristics

Generally, elevated O_3_ concentrations could induce typical visible injury and the occurrence and visible extent of the symptoms correlated positively with the concentration and duration of O_3_ exposure [31]. Foliar visible injury may have started to develop after experiencing a different-time O_3_ exposure for different plants, even though negative effects may have started to develop on a microscopic level earlier [32]. In this study, elevated O_3_ (160 ppb for two weeks) induced serious visible injury in *L. perenne* and *F. arundinacea* leaves, in agreement with our previous studies in leaves of *Trifolium repens* exposed to 80 ppb O_3_ for 5 days [33] and *Poa pratensis* (120 ppb O_3_ for 7 days) [34]. In addition, similar foliar O_3_ injuries were observed on crops such as *Medicago truncatula* exposed to 70 ppb O_3_ for 6 days [8], winter wheat to 120 ppb O_3_ for two months [10], rice cultivars 120 ppb O_3_ for one day [35]. However, we found that no foliar visible injury symptom was observed in plants exposed to 80 ppb O_3_. It might imply that plants probably maintained higher ability of self-repair or detoxification under the chronic fumigation by lower O_3_ level than that under the acute fumigation by elevated O_3_ concentration (160 ppb), which resulting into irreversible injury due to serious damage in ability of self-repair or detoxification. In other words, it could be only a matter of time before occurrence of foliar injury symptom in plants exposed to 80 ppb O_3_ in this study.

Actually, we found that elevated O_3_ concentrations (both 80 and 160 ppb) induced a significant reduction of growth with poor biomass regardless of plant species at the end of this experiment. The current study showed that the percentage of decrease in biomass at the end of this experiment was larger in *L. perenne* than in *F. arundinacea* exposed to 160 ppb O_3_, implying that the latter probably was more tolerant than the former. Similar studies found that elevated O_3_ concentrations decreased growth and inhibited biomass accumulation in herb species such as clovers [36,37], crops [6,11,38] and trees [3,12,39,40]. R/S ratio is usually used for assessing plant health, and its variation is a common stress response driven by different plant strategies for coping with environmental stress [41]. In this study, we found that R/S ratio was not significantly affected by elevated O_3_ concentrations, in accordance with some results in poplar [42,43] and spring wheat [44], but in disagreement with many studies with a reduction of R/S ratio in trees [32,45,46] and herb species [47]. It was well known elevated O_3_ concentration led to lower roots allocation relative to shoots and, thus, reduced R/S ratios. However, observed responses in R/S ratio are highly variable, which is associated with variations in experiments such as plant species or cultivars, growth and development stages, nutrient condition in soil and duration and level of gas fumigation [48].

### 3.2. Pigment Content and Chloroplast Ultrastructure Changes

Chlorophyll (Chl) is the most important pigment in plants and is usually embedded in the thylakoid membranes of chloroplasts [49,50]. Elevated O_3_ concentrations usually induced the reduction of photosynthetic pigments including Chl in plants. We found that 160 ppb O_3_ concentration in this experiment decreased total Chl and Car contents, in agreement with the result that elevated O_3_ concentration decreased total Chl and Car contents in maize leaves [51] and tree leaves [20]. However, no significant effect of 80 ppb O_3_ was found on the contents of pigment contents, as confirmed by no significant foliar injury symptom in appearance from the two plant species in this study. Similar results were found that chlorophyll content did not significantly vary among species or varieties under O_3_ exposure [52]. Actually, the effect of O_3_ on photosynthetic pigment contents depended on the time and dose of O_3_ exposure [53,54], as well plant development stage such as a study shown that elevated O_3_ did not affect chlorophyll content in young fully expanded leaves of *Betula pendula* saplings, but it reduced chlorophyll content in aging leaves [55].

It is well known that chloroplast is center of photosynthetic response and Chl synthesis. Under elevated O_3_ concentration, plant chloroplast structure could experience evident changes in micro-morphological characteristics, which might obviously impair their functionality and affect nutrient translocation [56]. In this study, we found that chloroplasts in O_3_-exposed leaves in the two plant species were abnormal and swollen in shape, and had many osmiophilic granules, indicating that an increase in osmiophilic granules could originate from the lipid-soluble degradation products from the thylakoid membranes under O_3_ stress [57]. In addition, chloroplasts showed greater damage and much accumulation of granules in *L. perenne* than in *F. arundinacea* exposed to the same O_3_ concentration, suggesting that the former could be more sensitive to O_3_ than the latter. Similar findings were observed that elevated O_3_ concentration strongly affected chloroplasts of urban tree species and induced a part or total deformation or disaggregation [20].

### 3.3. Oxidative Injury and Antioxidative Metabolism

As a strong oxidant, elevated O_3_ can cause membrane lipid peroxidation and induce oxidative injury of tissue cells in plants [58]. As the product of membrane lipid peroxidation, MDA content can increase in plants resulting from oxidative stress by elevated O_3_ concentrations [3,59]. In this study, we observed that elevated O_3_ concentrations significantly increased MDA level, suggesting the occurrences of membrane lipid peroxidation and oxidative stress after O_3_ exposure. This was in agreement with our previous study that elevated O_3_ concentrations significantly increased MDA content in herb species such as *T. repens*, *P. pratensis* and *F. arundinacea* [34], tree species such as *Quercus mongolica* and *Populus alba* and *Pinus tabulaeformis* [3,9,39]. Besides, exposure to elevated O_3_ concentrations often increased ROS production [60]. The two turfgrass species in the present study showed an increase in production rate of O_2_**·**^−^ and H_2_O_2_ content under elevated O_3_ concentrations. This indicated that the plants under elevated O_3_ exposure could enhance the risk of oxidative stress. Similar results were found that elevated O_3_ concentrations significantly increased H_2_O_2_ levels in leaves of Italian ryegrass (*L. multiflorum*) [14] and *Ginkgo bioloba* [61]. By contrast, H_2_O_2_ content in some plants decreased, which may be related to the increasing of scavenging ROS ability by high antioxidative level under O_3_ stress [10,12,54].

As we well known, antioxidant enzymes under O_3_ stress play an important role in preventing oxidative stress by scavenging ROS. A positive relationship has been found between antioxidant capacity in plants and tolerance to O_3_ [62]. In other word, the change of antioxidative enzyme activities and the extent of change determined the status of O_3_ stress and tolerance shown by species or varieties. In this study, we found that elevated O_3_ concentrations increased the activities of all the tested enzymes, in agreement with the findings in *T. repens* and *P. pratensis* [34] and peach tree cultivars [7], but in disagreement with the results found in *P. halepensis* [63], *Catalpa ovata* [64] and winter wheat [10] that elevated O_3_ decreased SOD and POD activities. Actually, the activities of some enzymes of most of plants showed the trend of increase first and then decrease with increasing of O_3_ exposure dose and duration [10,11,54], just like the change in APX activity from 80 ppb to 160 ppb O_3_ in this study. In the current study, SOD, CAT and POD activities showed higher level in leaves of *F. arundinacea* than *L. perenne* exposed to 160 ppb O_3_, indicating that the former could be more tolerant to O_3_ stress than the latter. For these enzyme activities, there was a significant interaction of O_3_ × species, indicating that different O_3_ effects between species. These different findings showed that species specific differences in antioxidant enzyme activities might reflect different physiological adjustment capacities in response to O_3_ stress [3]. However, the increases in the activities of antioxidative enzymes regardless of plant species in this study were not able to counteract membrane lipid peroxidation, cellular ultrastructural deterioration and damage, suggesting that the possible contribution of these physiological responses was insufficient to scavenge ROS and offset the oxidative stress induced by elevated O_3_ concentration in plants even though no visible injury was observed. Therefore, the damage extent by elevated O_3_ concentration might be underestimated only by evaluating foliar injury or chlorophyll content without considering the internal physiological changes further.

## 4. Materials and Methods

### 4.1. Study Site

This study was conducted in the Shenyang Arboretum (41°46′ N, 123°26′ E) of the Chinese Academy of Science (CAS) and closely located to a densely populated commercial center in Shenyang city of Northeast China, Liaoning Province. The arboretum with a mean elevation of 41 m and an area of 5 ha was founded in 1955, mainly planted with native tree species [65]. The study area belongs to warm temperate-zone semi-humid continental monsoon climate with an annual average temperature of 7.4 °C [66].

### 4.2. Experimental Design

This study was carried out in nine OTCs designed according to the design of [67] with three treatments: ambient air (AA, control, 40 ppb O_3_), two elevated O_3_ concentrations (80 ppb and 160 ppb O_3_). OTCs, 4 m in diameter and 3 m in height, were distributed randomly without mutual shading. Ozone was generated from O_3_ generator (Xinhang-2010, Shenyang, China) and then added to the OTCs. The generated O_3_ was directly dispensed to the open top chamber through a PVC pipe. The actual O_3_ concentrations in the chambers were monitored in real time and controlled for the targeted levels (80 ppb and 160 ppb) using an automated time-sharing system connected to an ozone analyzer (S-905, Auckland, New Zealand) and all data were stored using a data logger (CR800, Logan, UT, USA). More detailed information can be found in our previous experiments by [3,66].

### 4.3. Plant Materials and Treatments

Two cool-season turfrass species used in the experiment was perennial ryegrass (*L. perenne* cv. Lark) and tall fescue (*F. arundinacea* cv. Pixie), obtained for seeds from a northern company of China (Beijing Green Jinghua Ecological Landscape Co., Ltd., Beijing, China). The seeds were sown on 12 August 2015 into the plastic pots (20 cm in diameter, 15 cm in depth, 0.5 g per pot) filled with mixed loam and sand (3/1, *v*/*v*) and cultured in the greenhouse for 30 days, and then all the pots were divided into three groups to move into the OTCs with the treatment of AA and 80 ppb and 160 ppb O_3_, respectively. The plants were fumigated with elevated O_3_ concentrations for 9 h daily in the daytime (8:00–17:00). During the treatments, the positions of the pots in OTCs were randomly exchanged daily in order to minimize positional effects. The pots were often watered with enough tap water in order to keep them close to field capacity. After two weeks of gas fumigation, the older leaves were sampled for related physiological measurements and transmission electron microscopy analysis.

### 4.4. Measurements of Growth and Photosynthetic Pigment Contents

For growth, the different tissues (leaves, stems and roots) were separated and oven-dried to a constant dry weight at 60 °C to obtain dry biomass each part. Root/shoot (R/S) ratio was calculated by total dry root biomass per pot divided by total dry above-ground (shoot with leaves and stems) biomass per pot. Chlorophyll in leaves was extracted with 95% ethanol (*v*/*v*) in the dark for 72 h at 4 °C [20] and quantified spectrophotometrically (UV-1800, Shimadzu, Japan). Chlorophyll a (Chla) and b (Chlb) contents were measured and determined at wavelength of 649 and 665 nm. Carotenoids (Car) were measured at 470, 649 and 665 nm according to the modified methods of [68].

### 4.5. Evaluation of Oxidative Injury

Malondialdehyde (MDA) content was measured according to the method of [69]. The absorbance of thiobarbituric acid reactive substances in the reactive mixture was measured at 532 and 600 nm. The non-specific absorbance at 600 nm was subtracted from the absorbance at 532 nm. MDA content was calculated by an extinction coefficient of 155 (mM) ^−1^ cm^−1^.

Superoxide anion radical (O_2_**·**^−^) production rate was determined according to [70] with a slight modification. Samples of frozen leaves (0.2 g) were ground with liquid nitrogen and homogenized in 2 mL ice-cold 100 mM phosphate buffer (pH 7.8). The homogenates were centrifuged at 12,000× *g* for 30 min at 4 °C, and the supernatants were used for the analysis of O_2_**·**^−^ production rate. The supernatant was added to 10 mM hydroxylamine hydrochloride at 25 °C for 1 h, 17 mM p-aminophenic acid and 7 mM α-naphthylamine were added, and measured the absorbance at 530 nm after 20 min.

H_2_O_2_ content was measured as described by [71]. Leaf samples (0.2 g) were ground and centrifuged at 12,000× *g* for 20 min at 4 °C. The reaction solution was used to measure the H_2_O_2_ content at 415 nm by using UV-1800 spectrophotometer (Shimadzu, Kyoto, Japan).

### 4.6. Determination of Antioxidant Enzyme Activities

Leaf samples (0.2 g) were powdered with liquid nitrogen and ground with 5 mL 0.05 M phosphate buffer (pH = 7.8). The mixture was centrifuged at 12,000× *g* at 4 °C for 15 min, and the supernatant was used to analyze the activities of SOD, CAT, POD and APX. SOD activity was measured as described by [72]. The reaction system included 0.05 M phosphate buffer, 130 mM methionine, 630 µM NBT, enzyme extract and 13µM riboflavin. SOD activity was measured at 560 nm. One unit of SOD was defined as the amount of enzyme added by 50% inhibition of NBT reduction. CAT activity was measured as described by [73]. POD activity was measured as described by [74] with some modifications. APX activity was measured according to the method reported by [75].

### 4.7. Transmission Electron Microscopy Analysis

To observe the ultrastructure of chloroplasts, the healthy tissues of fresh leaves were cut from the middle part of the leaf to ensure uniformity of sample material. Small pieces (1 mm^2^) of leaves were taken for electron microscope analysis. Fragments of leaves were fixed in 3% glutaraldehyde in 0.2 M sodium phosphate buffer (pH 7.0) for 30 min at 4 °C. The more detailed process was described according to [76].

### 4.8. Statistical Analysis

One-way analysis of variance (ANOVA) was used to detect significant differences of each parameter between AA and elevated O_3_ concentration. The significant differences among the different factors including grass species and O_3_ concentrations were analyzed by the least significance differences (LSD) at 95% confidence level by using SPSS statistical software (PASW 18.0, Chicago, IL, USA). Differences among the different factors were considered significant at *p* < 0.05. The main effects and interactions of species and O_3_ on physiological parameters were evaluated by general linear model (GLM).

## 5. Conclusions

In the present study, growth, foliar visible and oxidative stress injuries, antioxidative enzyme activities and chloroplast ultrastructure of two common cool-season turfgrass species were investigated and analysed under the different elevated O_3_ concentrations. No visible foliar injury was observed in leaves of *L. perenne* and *F. arundinacea* exposed to 80 ppb O_3_, except for 160 ppb O_3_ resulting into the similar injury symptoms characterized as the diffuse chlorotic stripes and brown necrosis appeared in leaves. Regardless of grass species, photosynthetic pigment contents in leaves had not significant change under 80 ppb O_3_, but decreased significantly under 160 ppb O_3_. Chloroplasts in O_3_-exposed leaves were abnormal and swollen in appearance with lots of osmiophilic granules regardless of plant species. In 160-ppb O_3_ exposed leaves, chloroplasts were partly damaged with an irregular and slack grana thylakoid, especially for *L. perenne*. MDA content, O_2_**·**^−^ production rate and H_2_O_2_ content in leaves showed higher values under elevated O_3_ concentrations than those under ambient air, indicating that oxidative stress occurred under elevated O_3_ exposure. However, O_3_ fumigation increased significantly antioxidative enzyme activities in leaves of the two turfgrass species, and the increased enzyme activities in *F. arundinacea* leaves showed higher level than those in *L. perenne* leaves, suggesting that the former showed higher O_3_ stress adaptation than the latter. In addition, elevated O_3_ induced a significant reduction in root and total biomass of the two turgrass species. Based on comparative and comprehensive responses of the physiological characteristics of the two cool-season plant species, we found that *F. arundinacea* could be more tolerant to O_3_ than *L. perenne*. What’s more, the damage extent of elevated O_3_ concentration to the two grass species might be underestimated only by evaluating the changes in foliar injury or chlorophyll content regardless of the internal physiological metabolisms including chloroplast ultrastructure change and ROS accumulation of plants, particularly for the young seedlings under the short-term acute exposure of O_3_. Future studies should be required to examine the physiochemical changes by the long-term study under the duration of acute O_3_ stress, especially at the different stages of growth and development of cool-season plant species.

## Figures and Tables

**Figure 1 ijms-23-05153-f001:**
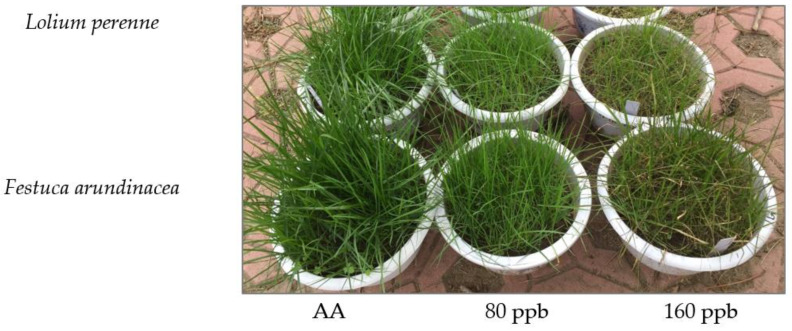
Effects of elevated O_3_ concentrations on foliar visible injury in *Lolium perenne* and *Festuca arundinacea*. No visible injury was observed under ambient air (AA) and 80 ppb O_3_, except for 160 ppb O_3_.

**Figure 2 ijms-23-05153-f002:**
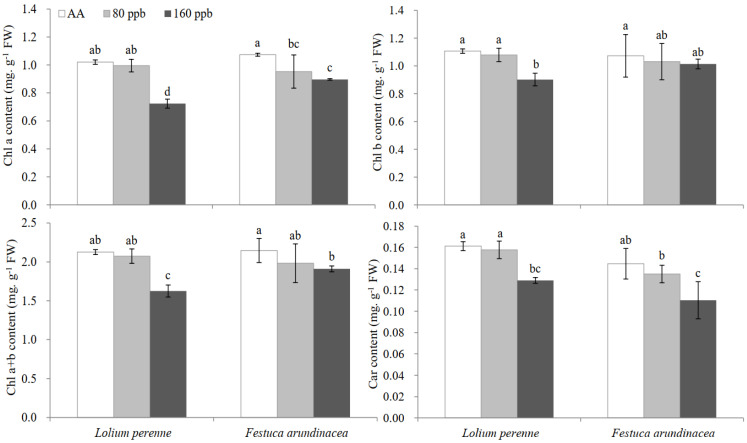
Effects of elevated O_3_ concentrations on chlorophyll a (Chl a), chlorophyll b (Chl b), Chl a + b and Carotenoids (Car) in leaves of two cool−season turfgrass species. Each value represents the average (±SE) of 3 replicates. Different lowercase letters within a row indicate significant differences among the different factors (*p* < 0.05).

**Figure 3 ijms-23-05153-f003:**
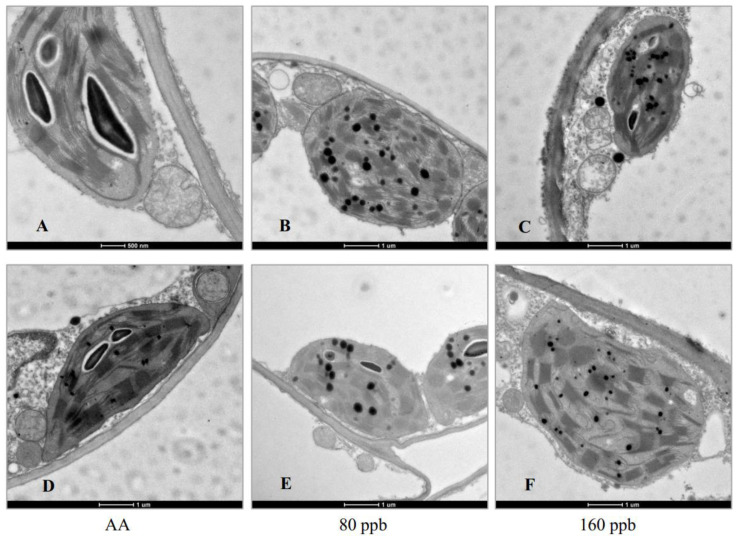
Transmission electron microscope (TEM) observation on the changes in ultrastucture of *Lolium perenne* and *Festuca arundinacea* leaves under elevated O_3_ concentrations for two weeks. (**A**–**C**) Details of TEM micrographs of cross sections of *L. perenne* leaves. (**D**–**F**) Details of TEM micrographs of cross sections of *F. arundinacea* leaves. (**A**,**D**) Chloroplasts under ambient air (AA) exhibited an elliptical shape and a typical regular lamellar grana structure consisting of thylakoid and several starch grains. (**B**,**E**) Chloroplasts under 80 ppb O_3_ exhibited a rounded and abnormal shape and a typical irregular lamellar grana structure consisting of thylakoid and numerous osmiophilic granules. (**C**,**F**) Chloroplasts under 160 ppb O_3_ showed a serious deformation in shape and were partly damaged with an irregular and slack grana thylakoid and completely separated from cell wall.

**Figure 4 ijms-23-05153-f004:**
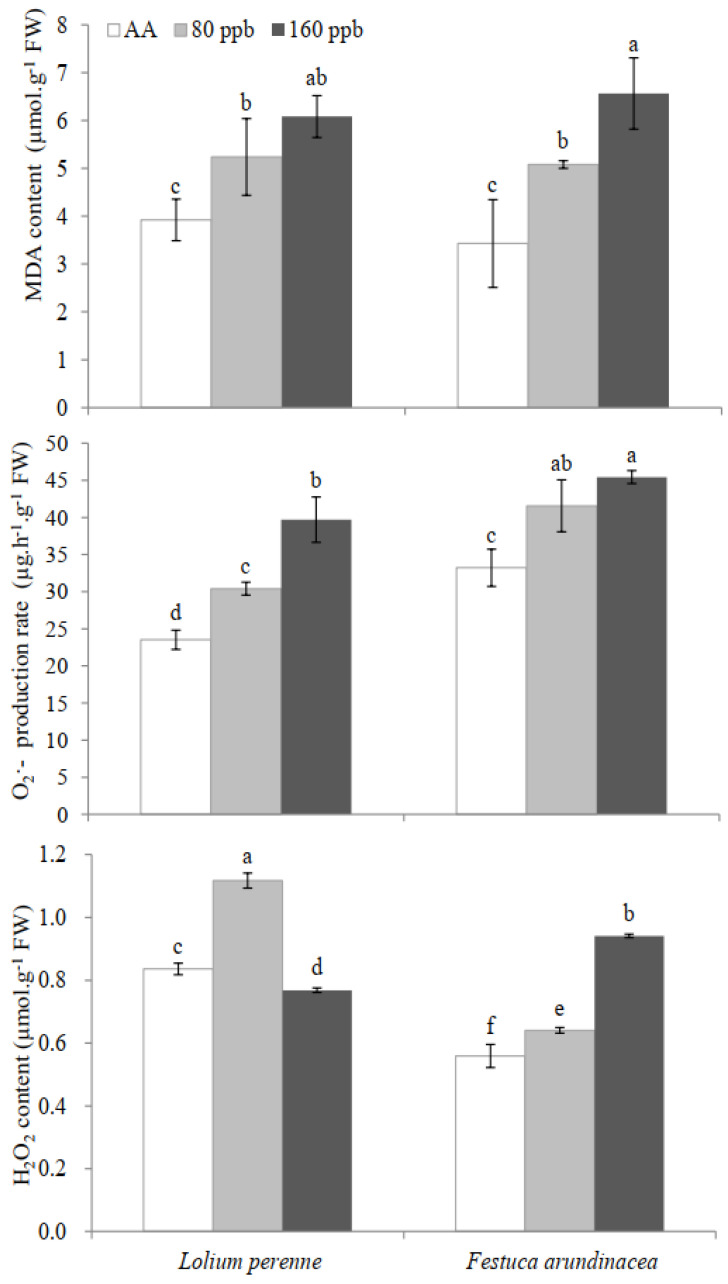
Effects of elevated O_3_ concentrations on malondialdehyde (MDA) content, superoxide anion radical (O_2_**·**^−^) production rate and hydrogen peroxide (H_2_O_2_) content in leaves of *Lolium perenne* and *Festuca arundinacea*. Each value represents the average (±SE) of 3 replicates. Different lowercase letters within a row indicate significant differences among the different factors (*p* < 0.05).

**Figure 5 ijms-23-05153-f005:**
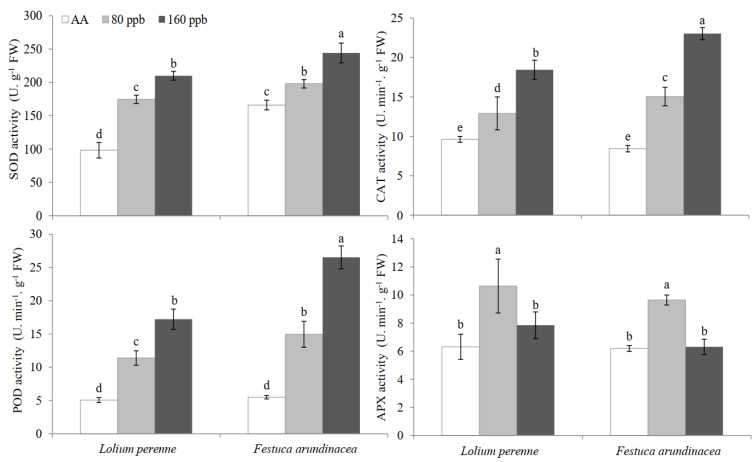
Effects of elevated O_3_ concentrations on the activities of superoxide dismutase (SOD), catalase (CAT), peroxidase (POD) and ascorbate peroxidase (APX) in leaves of *Lolium perenne* and *Festuca arundinacea*. Each value represents the average (±SE) of 3 replicates. Different lowercase letters within a row indicate significant differences among the different factors (*p* < 0.05).

**Figure 6 ijms-23-05153-f006:**
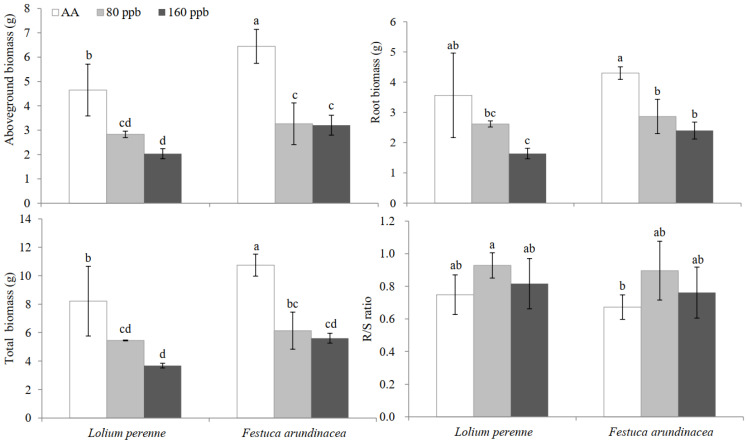
Effects of elevated O_3_ concentrations on aboveground biomass, root and total biomass and root/shoot (R/S) ratio of *Lolium perenne* and *Festuca arundinacea*. Each value represents the average (±SE) of 3 replicates. Different lowercase letters within a row indicate significant differences among the different factors (*p* < 0.05).

**Table 1 ijms-23-05153-t001:** ANOVA results (*p* values) for main effects and interactions of species and O_3_ on physiological parameters in *Lolium perenne* and *Festuca arundinacea*.

	Species	O_3_	Species × O_3_
Shoot biomass	**0.003**	**<0.001**	0.240
Root biomass	0.077	**0.001**	0.742
Total biomass	**0.010**	**<0.001**	0.418
R/S ratio	0.406	0.065	0.960
MDA	0.852	**<0.001**	0.430
O_2_^−^.	**<0.001**	**<0.001**	0.145
H_2_O_2_	**<0.001**	**<0.001**	**<0.001**
SOD	**<0.001**	**<0.001**	**0.004**
CAT	**0.006**	**<0.001**	**0.004**
POD	**<0.001**	**<0.001**	**<0.001**
APX	0.081	**<0.001**	0.473
Chl a	**0.033**	**<0.001**	**0.016**
Chl b	0.810	0.058	0.256
Chl a + b	0.270	**0.001**	0.074
Car	**0.002**	**<0.001**	0.875

MDA—Malondialdehyde, O_2_^−^.—Superoxide anion radical, H_2_O_2_—hydrogen peroxide, SOD—superoxide dismutase, CAT—catalase, POD—peroxidase, APX—ascorbate peroxidase, Chl—chlorophyll, Car—carotenoids. Significant effects (*p* < 0.05) are marked in bold.

## Data Availability

Data sharing is not applicable to this article.

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
