# Peer review of "Responses of Growth, Oxidative Injury and Chloroplast Ultrastructure in Leaves of Lolium perenne and Festuca arundinacea to Elevated O3 Concentrations"

_ijms, 2022, doi:10.3390/ijms23095153_

Round 1
Reviewer 1 Report
The authors have investigated the effect of elevated ozone on the growth, foliar visible and oxidative stress injuries, antioxidative enzyme activities and chloroplast ultrastructure of two common turf-type plant species. Between the two turf grass species, L. perenne could be more sensitive to high ozone exposure than F. arundinacea.
Overall Comments:
- The manuscript must be proof read by native English speaker
- Was there any attempt to study stomata distribution and general pattern was conducted between the two plant species?
- Conclusions must be rewritten to make to clear
Title:
- Please mention the name of the two turf grass species used the manuscript in the tile itself.
- Kindly rewrite it to highlight some of the interesting results based on the present study?
Abstract:
Page1, Line: 18; In place of writing the significant decrease plz mention the percentage change data
Page1, Line 23: Plz use only one of these term high or elevated or exposure throughout the manuscript.
Page 1, Line 23: Plz mention the results in percentage decrease or increase in the level of ant oxidative enzymes (superoxide dismutase, catalase and peroxidase, ascorbate peroxidase)
Page 1, Line 25: It was concluded from……….. ultra-structure and ROS accumulation.
The abstract needs to be rewritten so that the results based on the present study can be highlighted.
Introduction
Page1, Line 33: Please mention the rate (annual increase) also?
Page2, Line 46: “In particular, O3 entered the …………………….physiological reactions and stress responses” pLz check the grammar and the tense of the statement?
Page2, Line 54: “Actually, plants including herb ………………..hydrogen peroxide (H2O2) and superoxide anion (O2· –)”. Plz rewrite this to make it clearer?
Page2, Line 56: “Changes in the activities of ………….stress by scavenging ROS in many plants”. More citations needs to be reported here?
Page2, Line 62: “are two improved turf-type species in cool-season pants “Plz replace “pants” with “Plants”
Page 7 Line 171-172: Plz rewrite to make it clearer
- Discussion
Page 7, Line: 175-176 and Page 8 Line: 214-215 Actually, seems to contradictor? Plz recheck and rewrite to make it clearer?
- Materials and Methods
- Which leaves (either newly emerged or older leaves) were analysed for the photosynthetic pigment estimation
- What was the age of the plants when the sampling for biochemical analysis was done?
- Was there any differences in the growth pattern and life span of the two selected species in this study?
- Conclusions
- Line 408: “Photosynthetic pigment contents……under 160 ppb O3. Please write this sentence it has some grammatical issue.
- Line 422: “What’s more, O3 sensitivity and the damage ex- 422 tent might be underestimated only by evaluating foliar injury or chlorophyll content with- 423 out considering the internal physiological changes, especially in chloroplast ultrastructure 424 and ROS accumulation.” Please rewrite this sentence to make it clear.
Figure and Tables
Figure 1: Its will be easy to grasp what are the visible changes if any among the photographs of AA, 80 ppb and 160 ppb ozone treatment for both the species are shown all together?
Reviewer 2 Report
In this manuscript, the authors examined the effect of high O3 for 14 days on growth, oxidative injury, antioxidative enzyme activities, and chloroplast ultrastructure for the two cool-season turfgrass species, Lolium perenne and Festuca arundinacea. The authors showed that O3 exposure of 80 ppb induced no significant changes in foliar visible injury and photosynthetic pigment contents, but induced high accumulation of malondialdehyde and reactive oxygen species in leaves and damage in chloroplast ultrastructure in the two plant species. From these results, the authors concluded that O3 sensitivity and the damage extent could be underestimated only by evaluating foliar injury or chlorophyll content without considering the internal physiological changes. I think this is a well organized study that provides new insights into the evaluation of ozone sensitivity in plants. However, I noticed a few things and will comment on them below.
Implications of ozone exposure in the short term:
It would be easier for the reader to understand if you could add a little more explanation in the intro and interpretation in the discussion as to why you chose the short 14-day ozone treatment.
About Interspecies Comparisons:
Since there are no interspecific differences in the effects of ozone on growth in Table 2, and at 80 ppb, Lolium perenne has a greater reduction in total biomass than Festuca arundinacea. I was wondering if the 160 ppb response alone is sufficient to determine susceptibility (L216-218). In addition, I think it is difficult to make interspecific comparisons based only on the photograph in Figure 3 (L253-255).
Determination of ozone sensitivity (Just a comment):
If the decision is made after the end of the growth period, I believe that the decrease in biomass will allow us to determine ozone sensitivity. Is it correct that the main focus of this study is to determine ozone sensitivity in a short period of time during the growth period?
Specific comments
L73-75: Could you please add a few more explanations leading to this hypothesis? I think that is what you are thinking from the contents of L65-66, but it is difficult to understand in its current state.
L83: Could you please explain AA in the body of the text?
Figure 1. Could you please add a photo of the 80 ppb if possible.
Figure 2. Could you please correct the description "among the four treatments (L104-105)" in the description. I believe that the two species are probably mixed and analyzed between 6 treatments. (Same for Figures 4, 5, and 6.)
Table 1 It is easier to understand if the abbreviations are explained.
Figure 3. It would be easier for the reader to understand if you could indicate which part you are referring to with arrows and symbols. Is there any reason why the scale is different for A only?
L132-134: The 160 ppb value for H2O2 is decreasing, could you please state? Please also revise the text of L268-270. If possible, could you please add an explanation in the discussion as to why this is the case?
Could you please provide additional discussion on the decrease in APX at 160 ppb in Figure 5?
Please describe actual ozone concentration measurements, if any.
L334-: Could you please provide a citation for Chlorophyll analysis method, if any.
L395: Could you please add the explanation of EO and Cd.
